# Polyoxometalates Impact as Anticancer Agents

**DOI:** 10.3390/ijms24055043

**Published:** 2023-03-06

**Authors:** Fátima Carvalho, Manuel Aureliano

**Affiliations:** 1Faculdade de Medicina e Ciências Biomédicas (FMCB), Universidade do Algarve, Campus de Gambelas, 8005-139 Faro, Portugal; 2Faculdade de Ciências e Tecnologia (FCT), Universidade do Algarve, 8005-139 Faro, Portugal; 3Centro de Ciências do Mar (CCMar), Universidade do Algarve, 8005-139 Faro, Portugal

**Keywords:** polyoxometalates, polyoxovanadates, polyoxotungstates, cell viability, cell cycle, drugs, cancer

## Abstract

Polyoxometalates (POMs) are oxoanions of transition metal ions, such as V, Mo, W, Nb, and Pd, forming a variety of structures with a wide range of applications. Herein, we analyzed recent studies on the effects of polyoxometalates as anticancer agents, particularly their effects on the cell cycle. To this end, a literature search was carried out between March and June 2022, using the keywords “polyoxometalates” and “cell cycle”. The effects of POMs on selected cell lines can be diverse, such as their effects in the cell cycle, protein expression, mitochondrial effects, reactive oxygen species (ROS) production, cell death and cell viability. The present study focused on cell viability and cell cycle arrest. Cell viability was analyzed by dividing the POMs into sections according to the constituent compound, namely polyoxovanadates (POVs), polyoxomolybdates (POMos), polyoxopaladates (POPds) and polyoxotungstates (POTs). When comparing and sorting the IC_50_ values in ascending order, we obtained first POVs, then POTs, POPds and, finally, POMos. When comparing clinically approved drugs and POMs, better results of POMs in relation to drugs were observed in many cases, since the dose required to have an inhibitory concentration of 50% is 2 to 200 times less, depending on the POMs, highlighting that these compounds could become in the future an alternative to existing drugs in cancer therapy.

## 1. Introduction

The application of metals such as platin (Pt), lithium (Li), tungsten (W), gold (Au), and vanadium (V), among others, within chemical species, complexes, compounds and/or nanoparticles in biology has been a rapidly growing branch of science [1,2,3,4,5,6,7]. Besides platinum compounds, bio-active metal-based complexes, clusters such as gold compounds and polyoxometalates (POMs), and metal-based nanoparticles have shown anticancer, antiviral, and antibacterial activities, among others [1,2,3,4,5,6,7,8,9,10,11,12,13].

Regarding biomedical applications, the number of articles on POMs has tripled in the last decade [10], as illustrated by the well-studied polyoxovanadates (POVs) [14,15,16,17,18]. In fact, POMs are known to target several proteins such as aquaporins and P-type ATPases [11,13], although many other proteins and/or enzymes involved in many biochemical processes have been proposed to be affected [14,15,16,17,18,19]. The isopolyoxovanadate decavanadate (V_10_) is perhaps the most widely studied POV in biology, affecting several biochemical and cellular processes [14,15,16,17,18,19,20,21,22,23,24,25,26,27,28,29,30,31,32]. 

Polyoxometalates (POMs) are oxoanions of transition metal ions, such as V, Mo, W, Nb, and Pd, forming a variety of structures with a wide range of applications [10,12,15,16]. They may also include other elements in their structure such as P (phosphorus) or As (arsenic), among others, and may have one of the main metallic oxoanions absent and/or replaced by other metals, such as Co (cobalt) or Mn (manganese), resulting in versatile structures (Figure 1), which give them a wide variety of chemical and physical properties [21].

From the structure of a type of POM (polyoxometalate), such as Keggin or Dawson, (Figure 1D1–3,E1), it is possible to find lacunar derivatives, through the removal of one or more metal oxoanions. There are other compounds that can be formed from the structure of a type of Keggin POM, e.g., the sandwich type (Figure 1E2), which generally have two trilacunar anions separated by a belt of metallic cations.

In addition to the wide variety of chemical and physical properties, as mentioned before, this diversity of structures gives polyoxometalates numerous applications in several areas, such as environmental, chemical and industrial. Their effects are well known, mainly in catalysis, prevention of corrosion, and macromolecular crystallography, among others. Their usefulness in biomedicine [17,34,35] is also highlighted, namely through their antiviral [36,37,38] activity, where in the last two years, in the face of the world pandemic due to the SARS-CoV virus (severe acute respiratory syndrome coronavirus), studies have been carried out in an attempt to use POMs to help in fighting COVID-19 [39]. They are still useful in antitumor activities [40,41,42,43,44], and are antibacterial [16] and anti-inflammatory [45]. Additionally, they can also be used in biomedical engineering [46,47] in order to improve and develop innovative approaches to be applied in the prevention, diagnosis and therapy of diseases such as Alzheimer’s [48,49,50] and diabetes [51], where they tend to be demonstrated as potential drugs. In fact, the pharmacological action of POMs has sparked interest in them being potential candidates for therapeutic applications.

Since the beginning of the 21st century, and particularly recently, Aureliano’s research group and collaborators have been publishing review papers, editorials, chapters and regular papers regarding metal complexes and/or POMs biological functions and their applications [10,15,16,18,19,21,22,23,26,52,53,54,55,56]. Herein, POMs biological effects were further highlighted as possible anticancer agents in the near future. Among these effects, even though studies are scarce and the mechanisms of action are unclear, the cell cycle is particularly focused on. Still, POMs may thus be used as a potential strategy in the future as antitumor drugs with specific actions, which suggest the blockage of various cellular mechanisms such as the cell cycle.

## 2. Results and Discussion

### 2.1. Types of POMs

Of the selected articles, 18 compounds were found under study, belonging to four types of POMs, depending on their fundamental compound: POVs (polioxovanadates, polyoxometalates containing vanadium); those containing molybdenum (POMos, polyoxomolybdates); the POMs containing palladium (POPds, polyoxopaladates); and finally those containing tungsten (POTs, polyoxotungstates).

Starting with the POVs it was found that: (1) compounds with the molecular formula Na_4_Co_6_V_10_O_28_.18H_2_O (abbreviation CoV_10_), Na_3_(12H_2_O)H_3_V_10_O_28_.2H_2_O (abbreviation NaV_10_) an isopolyoxometalate with a decavanadate (V_10_) a well known type of structure [57]; (2) the POV with the formula K_12_[V_18_O_42_(H_2_O)]·6H_2_O (abbreviation V_18_) being a vanadium [58] with a Keggin structure; and (3) the compound 6{V_5_O_9_Cl(COO)_4_} (abbreviation VMOP-31) [59] with a Lindqvist’s type of structure.

Following the molybdenum compounds, we found the compound with the molecular formula K_2_Na[As^III^Mo_6_O_21_(O_2_CCH_2_NH_3_)_3_]·6H_2_O, also designated in the article by compound 1, in order to facilitate [60], better known as arsenomolybdate with an Anderson–Evans structure, involved in a silica nanosphere (K_2_Na[As^III^Mo_6_O_21_(O_2_CCH_2_NH_3_)_3_]·6H_2_O (abbreviation POM@SiO_2_) [61]; K_2_Na_2_[γ-Mo_8_O_26_(O_2_CCH_2_NH_3_)_2_]·6H_2_O, designated as compound 2, which are heteropolymolybdates [62]; with a Strandberg structure the [{4,4′-H_2_bpy}{4,4′-Hbpy}_2_{H_2_P_2_Mo_5_O_23_}].5H_2_O [63]; and finally, the [(Cu(pic)_2_)_2_(Mo_8_O_26_)]·8H_2_O [64].

There are also PONMs or polyoxopaladates, with the molecular formulas and respective abbreviations: Na_8_[Pd_13_As_8_O_34_(OH)_6_]·42H_2_O (abbreviation Pd_13_), Na_4_[SrPd_12_O_6_(OH)_3_(PhAsO_3_)_6_(OAc)_3_]_2_NaOAc·32H_2_O (abbreviation SrPd_12_), Na_6_[Pd_13_(PhAsO_3_)_8_]·23H_2_O (Pd_13_L), Na_12_[Sn^IV^O_8_Pd_12_(PO_4_)_8_]·43H_2_O (abbreviation SnPd_12_), Na_12_[Pb^IV^O_8_Pd_12_(PO_4_)_8_]·38H_2_O (abbreviation PbPd_12_) [65].

Lastly, the POMs that have the compound tungsten (POTs). We started with K_7_Na_3_[Cu_4_(H_2_O)_2_(PW_9_O_34_)_2_]20H_2_O (abbreviation PW_9_Cu) which has a Dawson structure [33]. Following a homochiral polyoxometalate {CoSb_6_O_4_(H_2_O)_3_[Co(hmta)SbW_8_O_31_]_3_}^15−^(1, hmta = hexamethylenetetramine) [66]; a tri-organic germanotungstate polyoxometalate-tin-substitute [67] {(n-Bu)Sn(OH)}_3_GeW_9_O_34_]^4−^·26H_2_O PAC-320 that has been shown to be an inhibitor of histone deacetylase (HDACi) [68] and has a Keggin structure. Finally, the POT {[Na(H_2_O)_4_][Na_0.7_Ni_5.3_(imi)_2_(Himi)(H_2_O)_2_(SbW_9_O_33_)_2_]} 10H_2_O, designated as compound 1 and H_3_[(CH_3_)_4_N]_4_[Na_0.7_Co_5.3_(imi)_2_(Himi)(H_2_O)_2_(SbW_9_O_33_)_2_] 12H_2_O, also known as compound 2 [67], whose structure is a hybrid Keggin sandwich.

The 18 compounds are represented below, in the form of a table, with the division of the POMs by main component and in the order in which they appear in the periodic table, i.e., vanadium (V); molybdenum (Mo); palladium (Pd) and tungsten (W), according to their increasing atomic number, as presented above. Each compound is presented in Table 1 with its designation, that is, its molecular formula or abbreviation when existing; the structure with the respective illustration; the type of structure; bibliographic reference and POM of the 13 selected articles (Table 1).

### 2.2. Types of Cancers

In these articles, several types of cancer were studied, and sometimes an article analyzed more than one type of cancer. These were namely gastric, colon, liver, lung, ovary, breast, prostate, leukemia, osteosarcoma, neuroblastoma and human hepatocellular carcinoma (Figure 2).

Through the analysis of Figure 2, it can be observed that the most studied types of cancer were lung and breast cancer, with six articles each, followed by liver cancer, covered in four articles.

### 2.3. POMs Effects

In the 13 articles, various effects of the selected POMs were analyzed, such as protein expression, mitochondrial effects, reactive oxygen species (ROS) production, cell cycle arrest, cell death, cell viability and anticancer activity, in vivo, or the antibacterial activity (Figure 3).

As seen in Figure 3, not all articles addressed all the effects of POMs, and there were effects addressed only in one article, such as antibacterial activity, analyzed in *Escherichia coli* (*E. coli*) regarding the effects of [(Cu(pic)_2_)_2_(Mo_8_O_26_)].8H_2_O, where it was shown to be quite effective with a minimum inhibitory concentration of approximately 135 μg/mL, which is the lowest value reported so far for any octamolybdate-based POM [64]. ROS production was also analyzed, as well as mitochondrial effects and protein expression in two, three and eight articles, respectively (Figure 3).

The production of ROS was verified in two articles, one of them with compounds containing noble metals (PONMs) [65]. The determination of the levels of superoxide ion produced was verified after 2 and 4 h of treatment. For this purpose, the dihydroethidium (DHE) method was used to detect cytosolic O_2_, which is not fluorescent but, being oxidized to O_2_^−^, emits fluorescence. Through the method presented above, it was observed that the polyoxopaladate Pd_13_ induced about 30% of increased production of superoxide ion (O_2_^−^) at 4h after treatment. Furthermore, the polyoxopaladates Pd_13_, SnPd_12_ and PbPd_12_ induced oxidative stress of HL-60 cells (human leukemia cells) resulting in an increase in the total production of reactive oxygen species [65].

Among the three articles selected for those that studied mitochondrial effects, the present study focused on the use of PAC-320. It was observed that with its use, in DU145 cells (human prostate tumor cells) there was a loss of MMP (mitochondrial membrane potential—ΔΨm). Consequently, cytochrome c is released in the cytosol, stimulating caspases that are associated apoptosis. Hence, it was deduced that apoptotic cell death was caused by the mitochondria-mediated pathway [68]. The same compound discussed above was one of the eight that referred to the study of protein expression. PAC-320 also inhibits the enzymatic activity of HDAC (histone deacetylases) 1, 2, 4, 5 and 6, but to a lesser extent HDAC 3 [68].

Only three articles performed in vivo studies in order to confirm or refute the anticancer activity of POMs. Of these three studies, *Mus musculus* (mice) were selected as an animal model. In one of these articles, nude mice with DU145 lineage (human prostate cancer cell line) were selected [68]. After this, these mice were divided into groups where each group was injected with a different compound. One group received a solution containing the compound PAC-320 (50 mg/kg) and another NaB (sodium butyrate) (1 g/kg). Others received SAHA (suberoylanilide acid, Vorinostat, an HDACi inhibitor, which was approved in October 2006 by the US Food and Drug Administration (FDA) for the treatment of cutaneous manifestations of T-cell lymphoma (CTCL) in patients with progressive, persistent or recurrent disease after two systemic therapies) [70], with a concentration of 40 mg/kg. All groups were injected daily for 16 days [68].

Through the results obtained, it was observed that the group treated with PAC-320 did not show a noticeable effect on body weight. In this group, the inhibition of the growth of prostate tumors was on average 69.2% compared to the control, treated with the vehicle alone (DMSO), while treatment with the drug SAHA or the compound NaB was inhibited by 55.5% and 64.2%, respectively. The weight of the tumors was reduced on the 17th day, when they were dissected [68]. To our knowledge, the precise mechanism of action responsible for the prostate tumor inhibition is yet to be clarified. In fact, although several putative mechanisms against cancer were recently reviewed [10,15,18], how the POM chemistry is specifically affecting the growth inhibition of prostate tumors needs further understanding.

In another study, over 12 consecutive days, mice with H22 liver tumor (Hepatoma-22, mouse liver tumor cell line) were administered treatments intraperitoneally, one with a saline solution (negative control group); cisplatin (CDDP or cis-diaminodichloroplatinum or cis-Pt) was used as the positive control group at a dose of 3 mg/kg. CDDP is a well-known platinum-based chemotherapy drug that is used to treat many types of cancer, including sarcomas, some carcinomas (e.g., small cell lung cancer and ovary cancer), lymphomas and germ cell tumors. Others received the studied compound, VMOP-31, at a dose of 12.5, 25 and 50 mg/kg [59]. On the 14th day, when they were excised, VMOP-31 led to a decrease in tumor weight compared to the saline group, but similar to that of the cisplatin group. Tumor inhibition rates of VMOP-31 at doses of 12.5, 25 and 50 mg/kg were determined to be 32.7, 39.6 and 56.4%, respectively [59]. Cisplatin has a tumor inhibition rate of 58.4%, which is comparable to the VMOP-31 tumor inhibition rate at 50 mg/kg. In this study, it was also observed that the body weight of the mice increased continuously, except for those in the cisplatin group, where the weight decreased continuously, which indicates that cisplatin has side effects [59].

Finally, we identified a study where 10 mice were randomly assigned and injected with mouse liver tumor cells (Hep-A-22) on their backs [57]. After 7 days of tumor cell administration, the mice were treated by intraperitoneal injections of CoV_10_ solutions with increasing concentrations (2, 6 and 12 mg/kg) [57], which were continued for 14 days. On the other hand, control mice were treated with saline solution for 2 weeks, under the same conditions that were used for animals treated with the CoV_10_ compound. As a result, it was observed that the average tumor weight with a concentration of 2 mg/kg with the CoV_10_ solution was 1.750 g [57], about 1.40-times lower than the average weight of the control group (2.440 g). As for the control group, compared with the value obtained (1.490 g) at a concentration of 12 mg/kg, a value about 1.6-times lower was observed. Comparing the control to a dose of 20 mg/kg of the approved drug, 5-Fu (fluorouracil is a widely used medicine in oncology, being therefore a base for a large part of current chemotherapy regimens to treat a wide spectrum of cancers), the mean tumor weight value was 0.477 g, i.e., 5.1-times lower. Note that here the dose used is higher than the dose used with the POM. Perhaps the higher dose of the drug justifies it being more harmful in reducing the weight of the mice, proving to be more toxic [57]. Furthermore, from the animal body weight data, it was found that CoV_10_ could decrease body weight less than the 5-Fu dose, thus showing less toxicity. It can also be seen that the inhibiting effect of the medium dose is better than that of the higher dosage (12 mg/kg). The reason may be that higher dosages of CoV_10_ can affect the function of immune organs, leading to decreased immune capacity [57].

However, several studies demonstrate that the oral administration of POMs is presumed safe and poses a low risk of potential health risks. Furthermore, for potential antidiabetic POTs it was concluded that the hepatotoxic and nephrotoxic effects could be considered as mild. Thus, besides POMs presenting higher antitumor activity and lower *toxicity* in in vitro and in vivo experiments, they have also been described as promising agents in the treatment of infectious diseases, diabetes and Alzheimer’s disease [69,71,72,73].

#### 2.3.1. Cell Viability

As discussed above, it was found that 13 articles analyzed cell death and that all selected articles refer to the effects of POMs on cell viability, and where cell cycle arrest was given (Figure 3). In these articles, the IC_50_ values (POM concentration that inhibits 50% of cell viability) of the various POMs selected in each article in the respective cell lines studied were analyzed. To understand which POM had the lowest IC_50_, a comparison was made, through a table containing the IC_50_ of the various compounds and the strains in which each one was applied, after exposure of 24, 48 and 72 h to POVs, POMos, POPds and POTs. The cell lines within each division were ordered alphabetically from A–Z (Table 2).

Five vanadium POMs were studied, from three POVs that were analyzed in eight cell lines (Table 2, Figure 4). Among these POVs, it was observed in the MCF-7 lineage (human breast cancer cell line) that the best IC_50_ was 0.53 µM at 72 h, achieved with the POV VMOP-31 [59] and corresponding to the best value of the polyoxovanadates. When comparing this value to V_18_ (45.95 µM) [58], in the same strain, whether at 24, 48 or 72 h, it always presents better results, being about 30 times (more potent) for 24 h. With CoV_10_, the SMMC-7721 lineage (human papillomavirus-related endocervical adenocarcinoma cell line) resulted in a value of <0.26 µg/mL [57], which proved to be about 73- to 196-times more efficient than NaV_1O_ compounds (18.90 µg/mL) and Co(Ac)2 (50.90 µg/mL) [57], such as in SK-OV-3 (human ovarian cancer cell line). For this strain, the best value is <0.24 µg/mL, when compared to the compounds NaV_10_ (9.56 µg/mL) and Co(Ac)2 (44.90 µg/mL), that is, CoV_10_ [57] was about 40- to 187-times more efficient than these compounds. These values cannot be compared with those of other vanadium compounds, since the measurement units are not the same.

In polyoxometalates containing molybdenum, six compounds were analyzed in six cell lines (Figure 4). It is observed in Table 2 that the A549 line (human lung cancer cell line) showed the best IC_50_ value to be 25 µM at 24 h, with the hybrid compound of structure Anderson–Evans (Figure 1G) and formula [(Cu(pic)_2_)_2_(Mo_8_O_26_)]·8H_2_O (Table 1). This same compound also showed a similar value (21.56 µM) in the HepG2 lineage (Human liver cancer cell line) at 48 h. Furthermore, in breast cancer, this compound showed a similar inhibition potency (24.24 µM) [71]. In the MCF-7 lineage, the compound K_2_Na[As^III^Mo_6_O_21_(O_2_CCH_2_NH_3_)_3_]·6H_2_O (POM@SiO_2_) with the same structure as the previous one, but inserted in silica nanoparticles (see structure in Table 1), presented the best IC_50_ value (1.70 µg/mL) at 72 h [61], however this cannot be compared, as they have different measurement units.

Of the five palladium-containing PONMs [65], all were studied in the same cell line SH-SY5Y (three times cloned subline from neuroblastoma cell line SK-N-SH (ATCC HTB-11). It was verified that the PONM that showed a better value both at 24 and 48 h was Pd_13_ (Table 1), with values of 7.20 and 4.40 µM, respectively, when compared to their hybrids containing phenyl groups (63.80 and 21.40 µM for Pd_13_L, 75.80 and 76.70 µM for SrPd_12_) [65], i.e., between 5 and 9 times for the first and between 11 and 17 times for the second, as the first was less efficient.

Regarding tungsten-containing compounds with effects on cell viability, eight compounds addressed in 15 different cell lines were found (Table 2, Figure 4). With POTs, in the A2780 and A2780cis lineage (human ovarian cancer cell line and the same lineage but resistant to cisplatin), the best values were obtained at 72 h with the compound [Co(H_2_O)_6_{CoSb_6_O_4_(H_2_O)_3_[Co(hmta)SbW_8_O_31_]_3_}]^13−^ which presents a homochiral, sandwich-like structure (Table 1), and were, respectively, 0.77 and 4.35 µM [66]. This was 6-times more efficient in the non-resistant cell line when compared with lines resistant to cisplatin treatment. In the AGS lineage (human stomach cancer cell line), the best value was 1.42 µM [67], for the hybrid compound that is a part of the sandwich, C_25_N_10_Na_0.7_Co_5.3_O_76_Sb_2_W_18_, and was the best result in this group of POMs.

On the other hand, on the BGC-823 cell line (human papillomavirus-related endocervical adenocarcinoma cell line), the best IC_50_ was found at 48 h with the pure inorganic compound {Sb_8_W_36_} [67]. For the transformed human embryonic kidney cell line (HEK293T), the hybrid C_25_N_10_Na_0.7_Co_5.3_O_76_Sb_2_W_18_ had the value of 103.09 µM at 48 h [67] and in the line OVCAR-3 (human ovarian cancer cell line), the best value was obtained at 72 h with POM [Co(H_2_O)_6_{CoSb_6_O_4_(H_2_O)_3_[Co(hmta)SbW_8_O_31_]_3_}]^13−^ being 1.78 µM [66]. Comparing the effects of tungsten-containing POMs on the AGS, BGC-823 and HEK293T cell lines, studied at 48h, it was found that the POTs are more efficient (in ascending order) in the AGS (1.42 µM), BGC-823 (8.68 µM) and HEK293T (103.09 µM) [67]. Thus, compound 2 is 73-times more potent in the transformed human embryonic kidney cell line when compared to the human stomach cancer cell line. Relative to the first POM referred to, at 72 h, a 2-fold greater efficacy was observed in the human ovarian cancer cell line A2780 (0.77 µM), compared to OVCAR-3 (1.78 µM) [66]. It was also found in the A549 lineage (human lung tumor lines) that the most effective compound was [Co(H_2_O)_6_{CoSb_6_O_4_(H_2_O)_3_[Co(hmta)SbW_8_O_31_]_3_}]^13−^ with a value of 12.75 µM [66] at 72 h, which was lower when compared to a value of 39.75 µM [67], found at 48 h, with the compound C_25_N_10_Na_1.7_Ni_5.3_O_82_Sb_2_W_18_. However, as previously observed in studies where they tested at 24, 48 and 72 h (increasing incubation time), the IC_50_ value tends to decrease.

In order to analyze whether antitumor drugs or POMs would be the most efficient, it was verified which drugs were tested in these strains and their respective IC_50_ values (Table 3). In Table 3, the lowest value among the approved drugs was 14.85 µM in the U937 strain, at 24 h, using the drug ATRA. While in relation to compounds, the lower value belongs to NaB, being 1.20 µM in the DU145 strain, at 72 h, which is about 14-times lower than the lowest value of the tested drugs. As for the cell viability of medically approved drugs or compounds, only cell lines of human origin were used. 

When comparing the cell viability of clinically approved drugs and POMs, it was observed, for example, a value of 49.79 µM using the drug MTX [60] (Table 3), and a value of 1.53 µM using the compound VMOP-31 [59], in the MCF-7 strain (breast cancer cell line) at 24 h (Table 2). It was verified that the IC_50_ value of the POM is lower, suggesting that the effects of POMs can overcome those of drugs. In fact, the dose necessary to have an inhibitory concentration of 50% is about 33-times lower than the dose that will be needed with a clinically approved drug.

Some drugs were tested in the strains mentioned above, such as ATRA (all-trans retinoid, also known as tretinoin), which is a drug used for the treatment of acne and acute promyelocytic leukemia (PML), having been tested in the HL-60 and U937 [60]. CDDP was tested on SH-SY5Y [65], MG-63 [33], AGS and BCG-823 [67] strains. The drug MTX (methotrexate), formerly known as amethopterin, a chemotherapeutic agent and immune system suppressor, used to treat cancer (cancer of the breast, leukemia, lung cancer, lymphoma, gestational trophoblastic disease and osteosarcoma), autoimmune diseases (i.e., psoriasis, rheumatoid arthritis and Crohn’s disease) and in ectopic pregnancy and for medical abortions, was analyzed here in the lines HepG2, A549 and MCF-7 [63].

It was found with the drug ATRA that the lowest IC_50_ value was 14.85 µM in the U937 strain, whereas a value of 20.76 µM was found in the HL-60 strain [60], both at 24 h. Cisplatin, at 24 h, is more effective in the SH-SY5Y strain (28.40 µM) [65], when compared to MG-63 (43.00 µM) [33]. On the other hand, at 48 h the lowest value was 5.78 µM in BCG-823 [67], conversely to the SH-SY5Y (11.60 µM) and AGS (17.44 µM) strains [67]. Finally, for the drug MTX at 24 h, the effect in the lines by order of increasing IC_50_ were A549 (26.93 µM), HepG2 (42.03 µM) and MCF-7 (49.79 µM) [63].

When comparing TSA (trichostatin A, an organic compound that serves as an antifungal antibiotic and selectively inhibits the histone deacetylase (HDAC) enzyme families of class I and II mammals) with NaB, in the LNCaP and DU145 strains at 72 h, it is concluded that in both strains the NaB compound has the advantage, showing values of 3.46 and 1.20 µM, compared to 98.14 and 59.45 µM [68] for TSA, for the same strains. It was also found that the antitumor activity of V_18_ was stronger than that of 5-Fu at 48 h for concentrations of 250 and 500 µM [58].

Moreover, taking in consideration the high IC50 values for POMs at normal cells compared to cancer cells, it is established that these compounds showed *high selectivity towards* the cancer cell lines [69]. Therefore, POMs selectively target *cancer cells* while sparing *healthy cells,* showing themselves to be promising agents in the treatment of cancer. In fact, POMs are expected to develop into the next generation of anticancer drugs [15,73].

#### 2.3.2. Effect of POMs on the Cell Cycle

Similarly to the cell viability discussed above, the effects of POMs on the cell cycle were also discussed in all articles selected (Figure 5). Particularly within the effects of compounds on the cell cycle, we wanted to highlight where each one interrupts the cycle. In Figure 5, the percentage of numbers of articles in which each phase of the cell cycle stagnated was summarized, referring to each POM and lines in which they were tested. Table 4 shows a better precision of the action of POMs in the arrest of the cell cycle of the different lineages studied. It was also verified that there are lines in which, depending on the POM used, the cell cycle phase where they stop can be different, such as the SMMC-7721 and MCD-7 cell lines, which are in the G2/M phase and in the S phase, respectively. Globally, 56% of the POMs stopped the cell cycle in the S phase (red), which is during DNA synthesis, whereas 36% blocked the G2/M phase (green), that is, when the transition from interphase to onset of mitosis occurs. While the G1 phase is characteristic of the maturation in proteins and RNA (ribonucleic acid) synthesis, it was still only 8% of the POMs, namely in the compound K_2_Na[As^III^Mo_6_O_21_(O_2_CCH_2_NH_3_)_3_]6H_2_O [60], represented with the color yellow, that arrested the cell cycle at this stage. It is also verified that different POMs can stop the cell cycle in several phases (Figure 5).

In order to analyze in detail which type of POM affected the different phases of the cell cycle in the respectively studied lines, a table was prepared, with the division of the phases of the cell cycle that were blocked by corresponding POMs and respective cell lines (Table 4).

It should be noted that in the articles where several POMs were studied, cell cycle arrest was only tested for the compound that showed the greatest efficacy. Likewise, of the cell lines analyzed, only those that had presented the best efficacy values were focused on, to verify cell arrest. Therefore, in Table 4, when in relation to Table 2, we verified a smaller number of POMs and cell lines analyzed, namely, three POVs in two cell lines and six POMos in four lines, with the majority being the study of the A549 line, human lung cancer cell line. There were only three POPds for the neuroblastoma line (SH-SY5Y), and finally we found four POTs for seven cell lines, all distinct from each other (Table 4). As estimated, in these studies the concentrations at which compounds have an effect on the cell cycle are close or below the IC_50_ previously determined and are described in Table 2.

As can be seen, the S phase is predominant under the action of the POMs with 14 of them, followed by the G2/M phase with nine polyoxometalates, and two in the G1 phase. It is also verified that VMOP-31 is the only one that has the ability to stop the cell cycle of SMMC-77221 cells in two distinct cell phases, namely S and G2/M [59]. On the other hand, it is also clear that the breast cancer cell line (MCF-7) on several compounds has its arrest in the G2/M phase, with the exception of the molybdenum compound involved in silica nanoparticles (K_2_Na[As^III^Mo_6_O_21_(O_2_CCH_2_NH_3_ )_3_]·6H_2_O (POM@SiO_2_)), stopping in the S phase [61].

Most of the proposed modes of action for antitumor POMs were recently reviewed [15]. Among these mechanisms, it was shown that POMs are able to affect DNA by interacting directly with it [15,58]. POMs were also suggested to cleave the phosphodiester bond and to affect DNA synthesis [15,74]. However, to our knowledge the processes responsible for the POMs effects that cause cell cycle arrest, as well as the effects in the cell cycle checkpoints, are unknown. For instance, the precise mechanism of action responsible for the POMs effects in DNA synthesis and/or in mitosis during the cell cycle process still needs to be deduced and needs further clarification.

## 3. Conclusions

In this review paper, bibliographic research was carried out with the keywords “polyoxometalates” AND “cell cycle”. Thirteen articles were selected on the effect of POMs with anticancer activities, namely in gastric, colon, liver, lung, ovary, breast, prostate, leukemia, osteosarcoma, neuroblastoma and human hepatocellular carcinoma. The types of cancer in which the articles most focused on were lung and breast cancer, with six articles each, followed by liver cancer, covered in four articles. The effects of POMs on cancer cells can be diverse, such as their interaction in the cell cycle, protein expression, mitochondrial effects, ROS production and cell viability. In the present study, we focused mainly on cell viability and cell cycle arrest, since all selected articles analyzed these two effects.

Cell viability was analyzed by dividing the POMs into sections according to the constituent compound, namely POVs, POMos, POPds and POTs. When the IC_50_ values were compared and sorted in ascending order, it was found that the POV (VMOP-31) at 24, 48 and 72 h presented, respectively, the lowest values of 1.52, 0.63, and 0.53 µM, in the MCF-7 (human breast cancer cell line). In clinically approved drugs, the lowest IC_50_ value was found to be 5.78 µM in the BCG-823 lineage at 48h using cisplatin. When comparing drugs and POMs, better results of POMs were observed. In many cases, the POMs dose required to have an inhibitory concentration of 50% is 2- to 200-times lower than the dose that would be necessary with a clinically approved drug. Therefore, POMs are future potential candidates for cancer therapeutic applications.

In addition to the cell viability, the effect of POMs on cell cycle arrest is highlighted. For the majority of the POMs, cell cycle arrest occurs mostly in the S phase (56%), where DNA synthesis occurs, whereas 36% blocks the G2/M phase. Fewer POMs interfere with the G1 phase (8%), that is, at the beginning of the cell cycle. Although the mechanism of action, directly and/or indirectly, responsible for the POMs cell cycle arrest is still to be deduced and clarified, the scientific evidence described above strengthens the potential use of such metallodrugs in anticancer therapy in the near future.

## Figures and Tables

**Figure 1 ijms-24-05043-f001:**
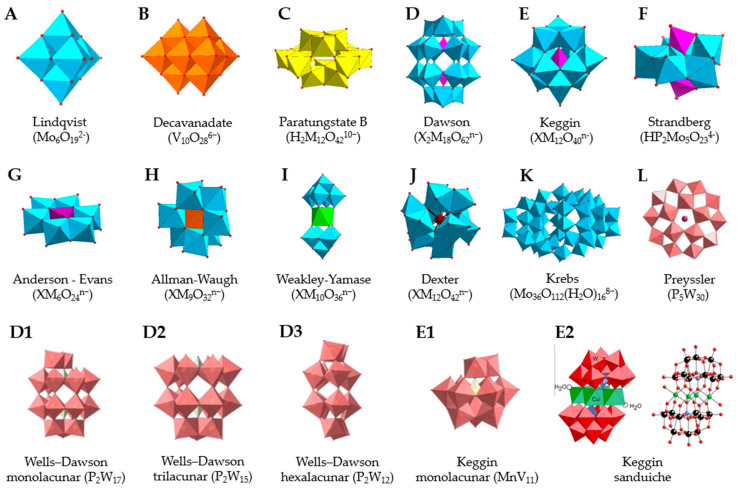
Examples of POM structures. (**A**)—Lindqvist (Mo_6_O_19_^2−^); (**B**)—Isopolyoxometalate (V_10_O_28_^6−^); (**C**)—Paratungstate B (H_2_M_12_O_42_^10−^); (**D**)—Dawson (X_2_M_18_O_62_^n−^); (**E**)—Keggin (XM_12_O_40_^n−^); (**F**)—Strandberg (HP_2_Mo_5_O_23_^4−^); (**G**)—Anderson–Evans (XM_6_O_24_^n−^); (**H**)—Allman–Waugh (XM_9_O_32_^n−^); (**I**)—Weakley–Yamase (XM_10_O_36_^n−^); (**J**)—Dexter (XM_12_O_42_^n−^); (**K**)—Krebs (Mo_36_O_112_(H_2_O)_16_^8−^); (**L**)—Preyssler (P_5_W_30_) [21]. Lacunar POM-like structures, Dawson: (**D1**)—Wells–Dawson Monocular (P_2_W_17_) [21]; (**D2**)—Wells–Dawson trilacunar (P_2_W_15_) [21]; (**D3**)—Wells–Dawson hexalacunar (P_2_W_12_) [21] and Keggin type: (**E1**)—Keggin monolacular (MnV_11_) [21] and (**E2**)—Keggin sandwich [33]. Adapted with copyright permission from MDPI and Elsevier, respectively from references [21,33].

**Figure 2 ijms-24-05043-f002:**
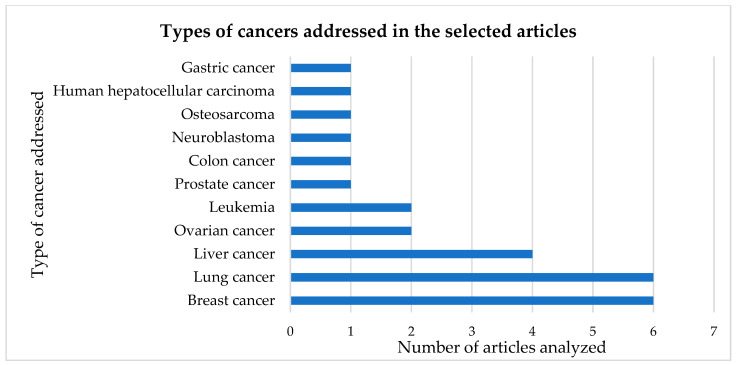
Number of articles for the different types of cancers studied.

**Figure 3 ijms-24-05043-f003:**
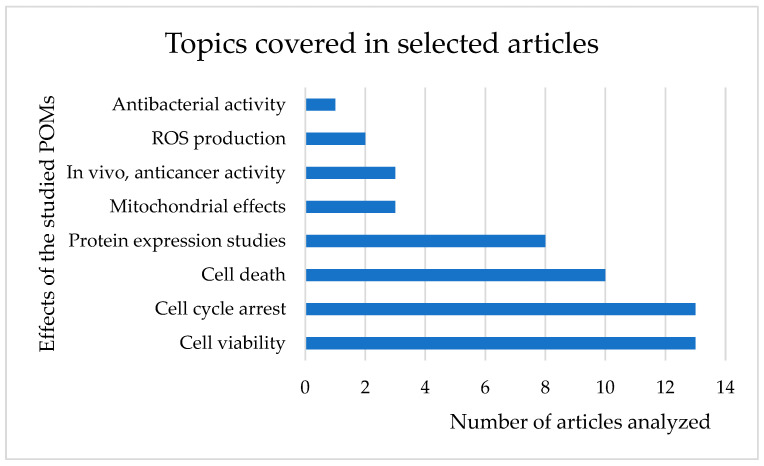
Number of articles in which each effect of the selected POMs was analyzed.

**Figure 4 ijms-24-05043-f004:**
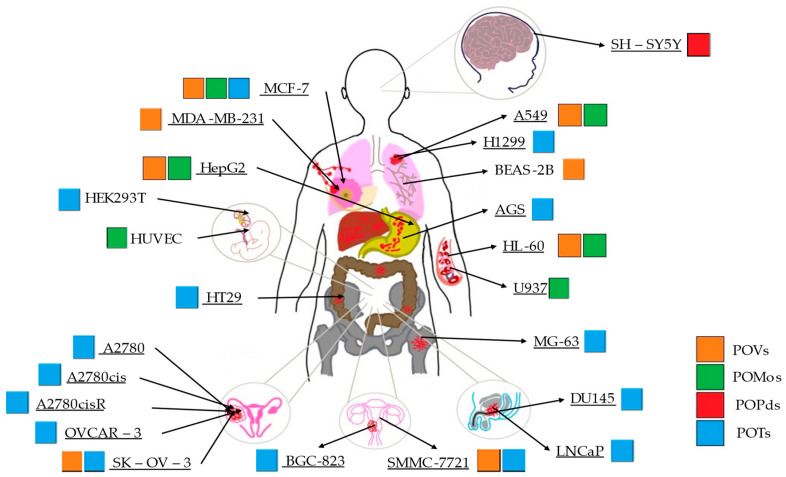
Illustration of the 23 human cell lines tested in the study of cell viability of different types of POMs. A total of 20 tumor lineages are underlined and 3 are not underlined, which are non-tumor cell lines. The color scheme is representative of each type of POM studied: POVs in orange, POMos in green, POPds in red and POTs in blue.

**Figure 5 ijms-24-05043-f005:**
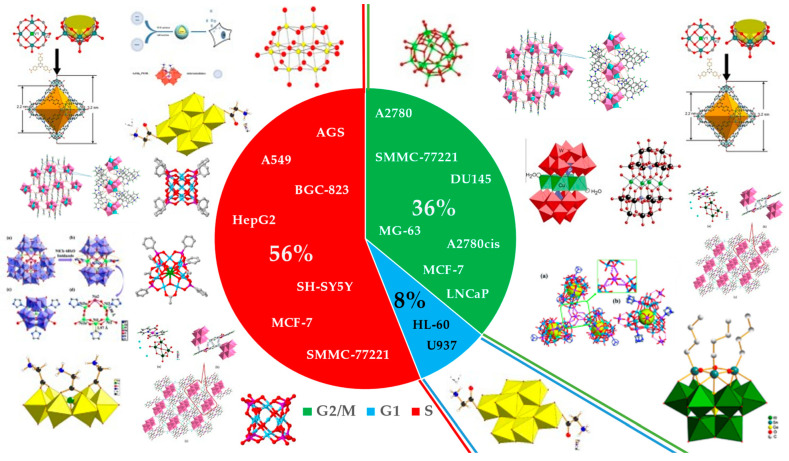
Percentage of the number of articles in which each cell cycle arrest occurred, referring to each POM used and respective cell lines. Color code: red—S phase [57,59,61,62,63,64,65,67]; green—G2/M [33,58,59,63,64,66,68] phase; and blue—G1 phase [60]. Reproduced from Refs. [63,66] with permission from the Royal Society of Chemistry. Reproduced from Refs. [33,57,60,62,71] with permission from Elsevier. Reproduced from Ref. [59] with permission from Wiley. Reproduced from Ref. [61] with permission from Plosone. Reproduced from Ref. [64] with permission from American Chemical Society. Reproduced from Ref. [65] with permission from Springer. Reproduced from Ref. [66] with permission from the Royal Society of Chemistry.

**Table 1 ijms-24-05043-t001:** Representation of the formula, structure and type of the POMs referred to in the selected articles.

POM Formula (Abbreviation)	Structure	Structure Type	Ref.
POVs			
Na_4_Co_6_V_10_O_28_.18H_2_O (CoV_10_) Na_3_(12H_2_O)H_3_V_10_O_28_.2H_2_O (NaV_10_)	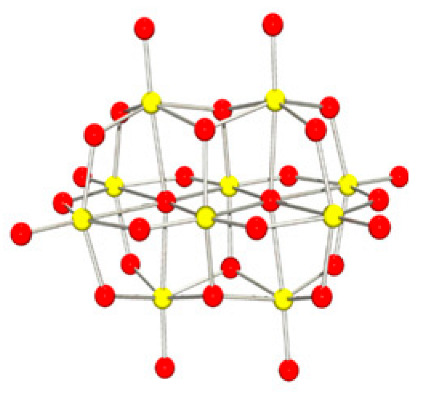	Decavanadate	[23,57]
K_12_[V_18_O_42_(H_2_O)] 6H_2_O (V_18_)	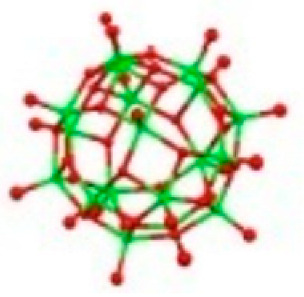	Keggin	[58]
6{V_5_O_9_Cl(COO)_4_} (VMOP-31)	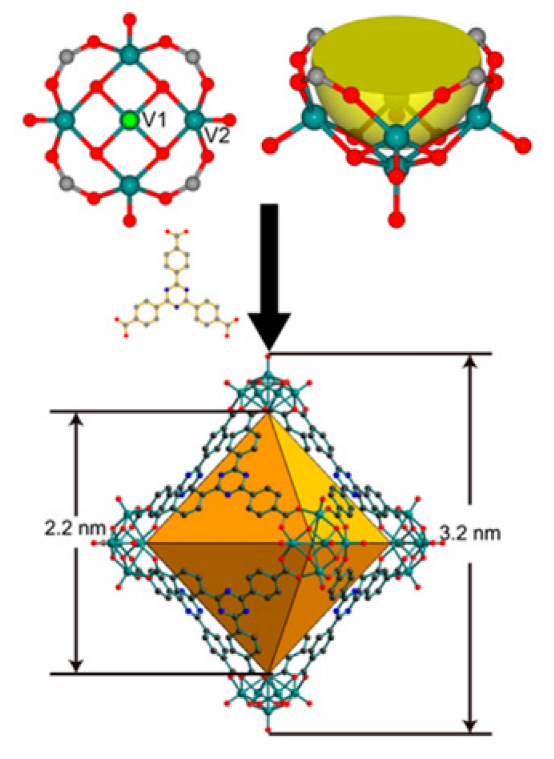	Lindqvist	[59]
POMos			
K_2_Na[As^III^Mo_6_O_21_(O_2_CCH_2_NH_3_)_3_] 6H_2_O—(compound 1)	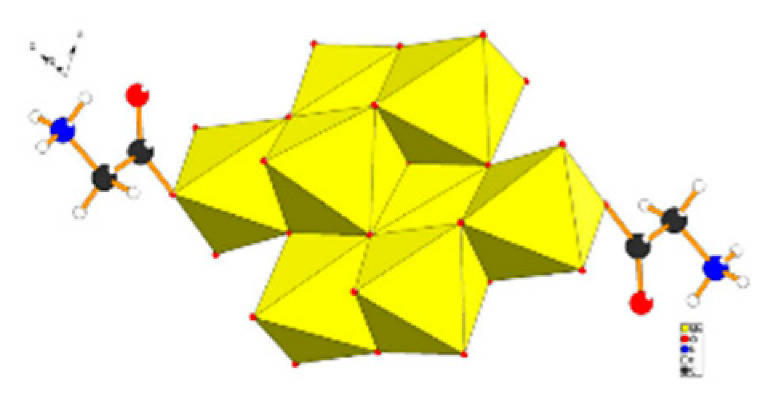	Anderson–Evans	[62]
K_2_Na_2_[γ-Mo_8_O_26_(O_2_CCH_2_NH_3_)_2_] 6H_2_O—(compound 2)	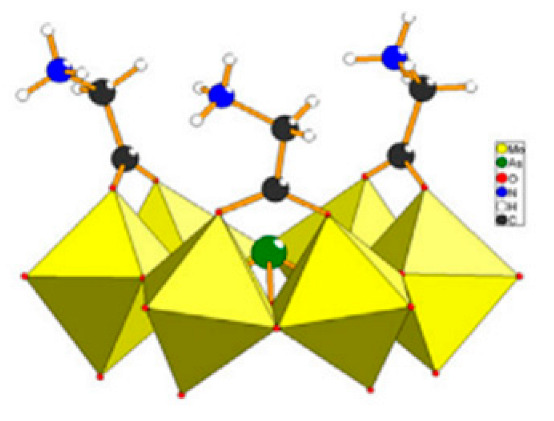		
[{4,4′-H_2_bpy}{4,4′-Hbpy}_2_{H_2_P_2_Mo_5_O_23_}].5H_2_O	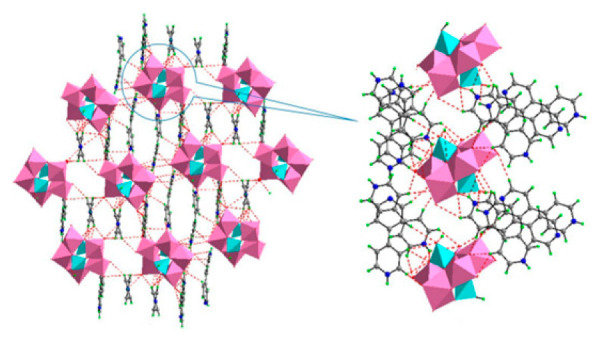	Strandberg	[63]
[(Cu(pic)_2_)_2_(Mo_8_O_26_)]·8H_2_O	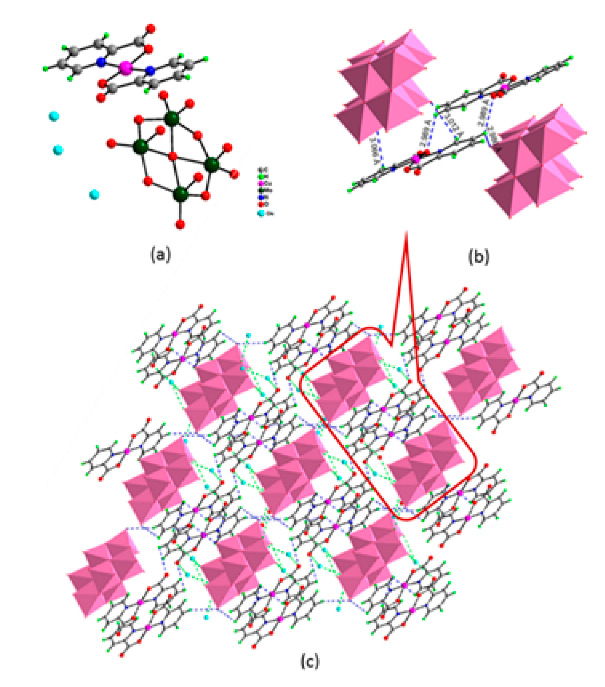	Anderson–Evans	[64]
K_2_Na[As^III^Mo_6_O_21_(O_2_CCH_2_NH_3_)_3_]·6H_2_O (POM@SiO_2_)	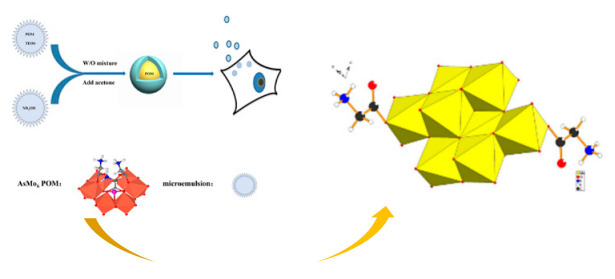	Anderson–Evans	[61,62]
K_2_Na[As^III^Mo_6_O_21_(O_2_CCH_2_NH_3_)_3_]·6H_2_O	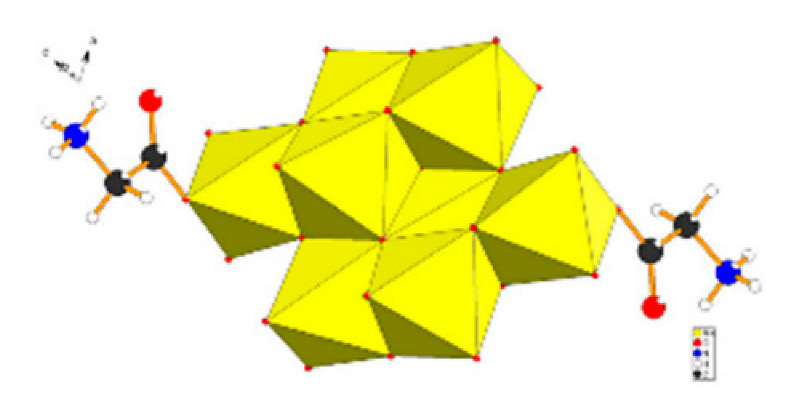	Anderson–Evans	[60,62]
POPds			
Na_8_[Pd_13_As_8_O_34_(OH)_6_]^.^42H_2_O (Pd_13_)	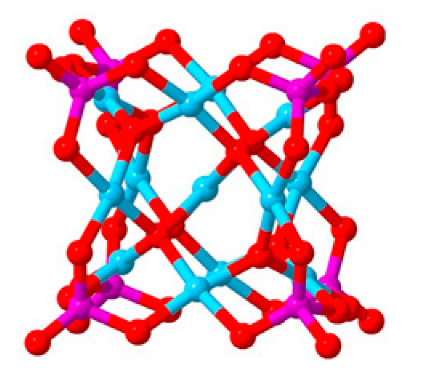	Unidentified	[65]
Na_4_[SrPd_12_O_6_(OH)_3_(PhAsO_3_)_6_(OAc)_3_] 2NaOAc·32H_2_O (SrPd_12_)	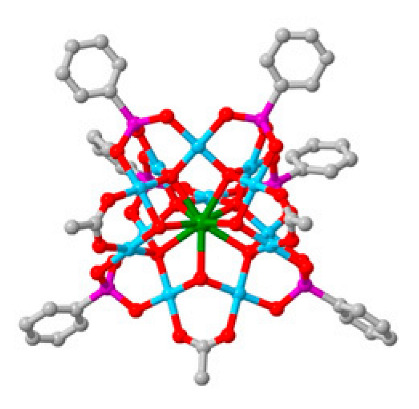		
Na_6_[Pd_13_(PhAsO_3_)_8_]·23H_2_O (Pd_13_L)	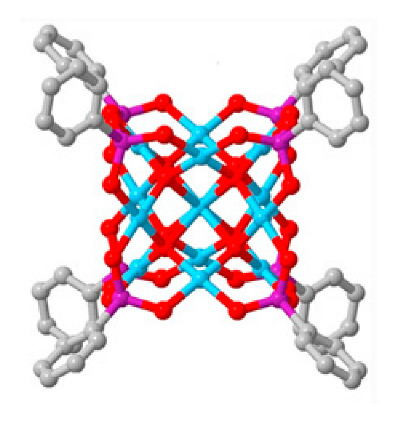		
Na_12_[Sn^IV^O_8_Pd_12_(PO_4_)_8_]·43H_2_O (SnPd_12_)	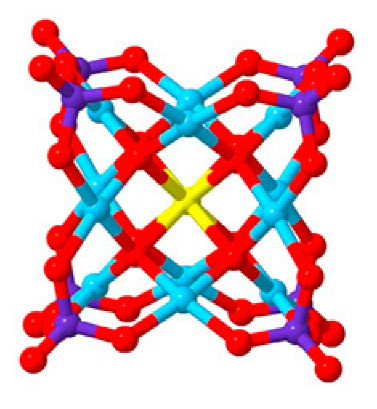		
Na_12_[Pb^IV^O_8_Pd_12_(PO_4_)_8_]·38H_2_O (PbPd_12_)	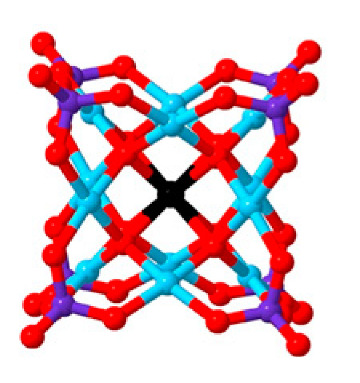		
POTs			
K_7_Na_3_[Cu_4_(H_2_O)_2_(PW_9_0_34_)_2_]20H_2_O (PW_9_Cu)	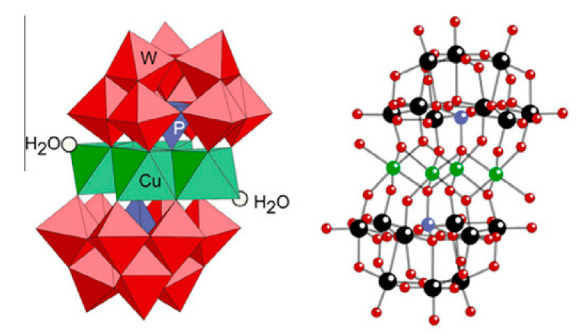	Keggin sandwich	[33]
{CoSb_6_O_4_(H_2_O)_3_[Co(hmta)SbW_8_O_31_]_3_}^15^	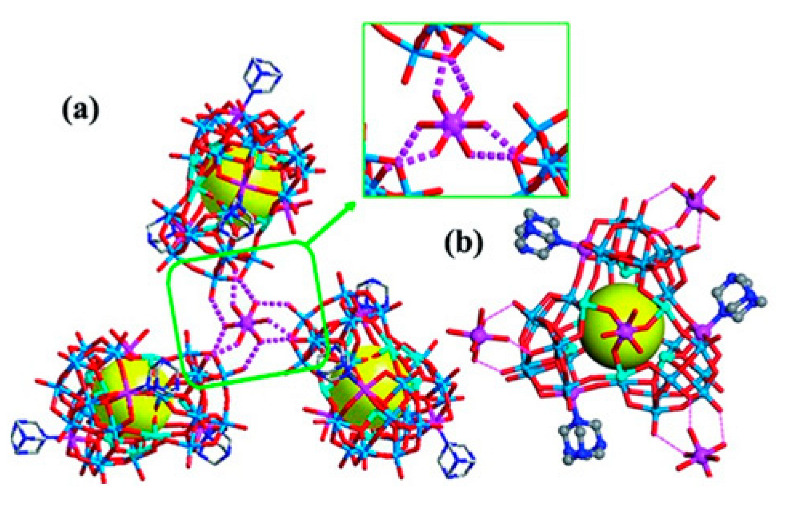	Homochiral	[66]
{(n-Bu)Sn(OH)}_3_GeW_9_O_34_]^4−^·26H_2_O (PAC-320)	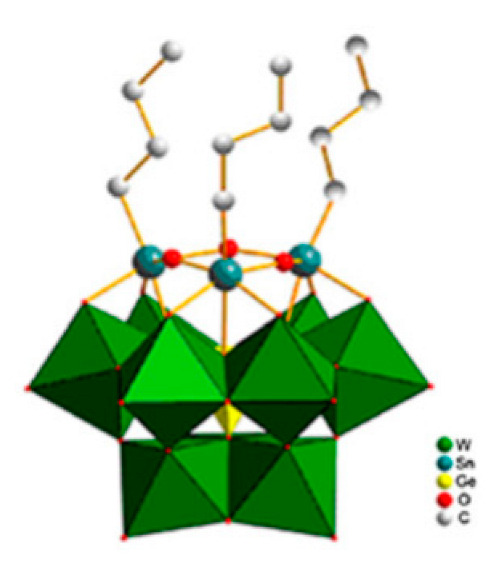	Keggin	[68,69]
H_2_[(CH_3_)4N]^4−^ {[Na(H_2_O)_4_][Na_0.7_Ni_5.3_(imi)2(Himi)(H_2_O)_2_(SbW_9_O_33_)_2_]} 10H_2_O (C_25_N_10_Na_1.7_Ni_5.3_O_82_Sb_2_W_18_) H_3_[(CH_3_)4N]_4_[Na0.7Co_5.3_(imi)2(Himi)(H_2_O)_2_(SbW_9_O_33_)_2_] 12H_2_O (C_25_N_10_Na_0.7_Co_5.3_O76Sb_2_W_18_)	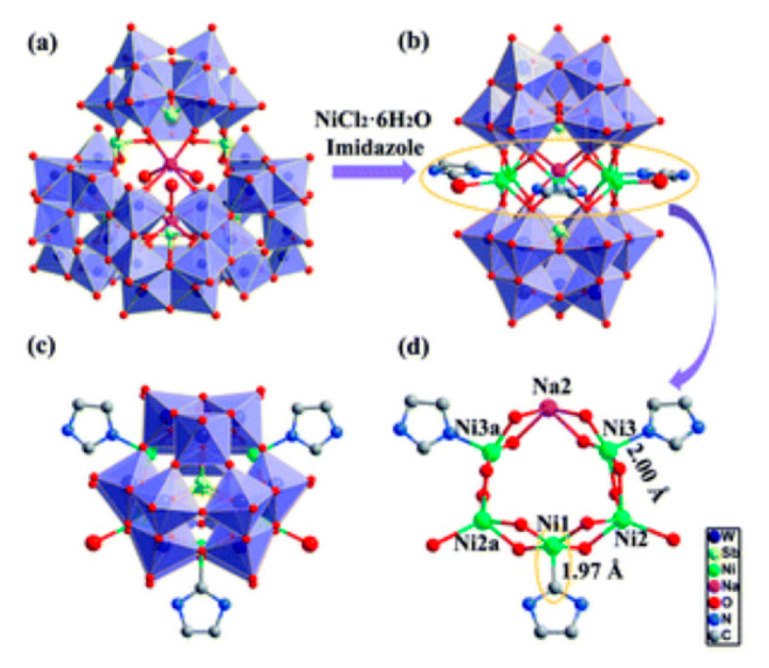	Keggin sandwich	[67]

**Table 2 ijms-24-05043-t002:** IC_50_ values (µM) of the different POMs studied in the respective cell lines.

Formula or Abbreviation	Cell Line	IC_50_ (µM)			Ref.
		24 h	48 h	72 h	
POVs					
VMOP-31	A549	2.64	0.74	0.55	[59]
	BEAS-2B	15.46	3.13	1.84	
	HL-60	14.66	2.04	1.61	
V_18_	MCF-7	45.95	11.95	12.49	[58]
VMOP-31	MCF-7	1.52	0.63	0.53	[59]
V_18_	MDA-MB-231	>500	360.32	135.66	[58]
CoV_10_	SK-OV-3			<0.24 *	[57]
NaV_10_	SK-OV-3			18.90 *	
Co(Ac)_2_	SK-OV-3			50.90 *	
CoV_10_	SMMC-7721			<0.26 *	
NaV_10_	SMMC-7721			9.56 *	
Co(Ac)_2_	SMMC-7721			44.90 *	
VMOP-31	SMMC-7721	2.66	1.17	0.75	[59]
POMos					
K_2_Na[As^III^Mo_6_O_21_(O_2_CCH_2_NH_3_)_3_] 6H_2_O	A549	332.87		180.40	[62]
K_2_Na_2_[γ-Mo_8_O_26_(O_2_CCH_2_NH_3_)_2_]	A549	1072.02		330.18	
[{4,4′-H_2_bpy}{4,4′-Hbpy}_2_{H_2_P_2_Mo_5_O_23_}].5H_2_O	4549	25.17			[63]
[(Cu(pic)_2_)_2_(Mo_8_O_26_)] 8H_2_O	A549		25.00		[64]
[{4,4′-H_2_bpy}{4,4′-Hbpy}_2_{H_2_P_2_Mo_5_O_23_}].5H_2_O	HepG2	33.79			[63]
[(Cu(pic)_2_)_2_(Mo_8_O_26_)]·8H_2_O	HepG2		21.56		[64]
K_2_Na[As^III^Mo_6_O_21_(O_2_CCH_2_NH_3_)_3_] 6H_2_O	HL-60	8.61	0.13	19.49	[60]
	HUVEC	889.18			
K_2_Na[As^III^Mo_6_O_21_(O_2_CCH_2_NH_3_)_3_]·6H_2_O (POM@SiO_2_)	MCF-7	40.00 *	10.80 *	1.70 *	[61]
[{4,4′-H_2_bpy}{4,4′-Hbpy}_2_{H_2_P_2_Mo_5_O_23_}].5H_2_O	MCF-7	32.11			[63]
[(Cu(pic)_2_)_2_(Mo_8_O_26_)] ·8H_2_O	MCF-7		24.24		[64]
K_2_Na[As^III^Mo_6_O_21_(O_2_CCH_2_NH_3_)_3_] 6H_2_O	U937	14.50	5.65	2.73	[60]
POPds					
PbPd_12_	SH-SY5Y	>>100	>>100		[65]
Pd_13_	SH-SY5Y	7.20	4.40		
Pd_13_L	SH-SY5Y	63.80	21.40		
SnPd_12_	SH-SY5Y	>>100	>>100		
SrPd_12_	SH-SY5Y	75.80	76.70		
POTs					
[Co(H_2_O)_6_{CoSb_6_O_4_(H_2_O)_3_[Co(hmta)SbW_8_O_31_]_3_}]^13−^	A2780			0.77	[66]
{Sb_9_W_21_}	A2780			4.44	
[Co(H_2_O)_6_{CoSb_6_O_4_(H_2_O)_3_[Co(hmta)SbW_8_O_31_]3}]^13−^	A2780cis			4.35	
{Sb_9_W_21_}	A2780cis			29.02	
[Co(H_2_O)_6_{CoSb_6_O_4_(H_2_O)_3_[Co(hmta)SbW_8_O_31_]_3_}]^13−^	A549			12.65	
C_25_N_10_Na_1.7_Ni_5.3_O_82_Sb_2_W_18_	A549		39.75		[67]
C_25_N_10_Na_1.7_Ni_5.3_O_82_Sb_2_W_18_	AGS		1.75		
C_25_N_10_Na_0.7_Ni_5.3_O_76_Sb_2_W_18_	AGS		1.42		
{Sb_8_W_36_}	AGS		2.86		
{SbW_9_}	AGS		26.22		
C_25_N_10_Na_1.7_Ni_5.3_O_82_Sb_2_W_18_	BGC-823		22.27		
C_25_N_10_Na_0.7_Ni_5.3_O_76_Sb_2_W_18_	BGC-823		20.51		
{Sb_8_W_36_}	BGC-823		8.68		
{SbW_9_}	BGC-823		188.28		
[Co(H_2_O)_6_{CoSb_6_O_4_(H_2_O)_3_[Co(hmta)SbW_8_O_31_]_3_}]^13−^	CT26			14.72	[66]
PAC-320	DU145			4.55	[68]
[Co(H_2_O)_6_{CoSb_6_O_4_(H_2_O)_3_[Co(hmta)SbW_8_O_31_]_3_}]^13−^	HT29			15.60	[66]
C_25_N_10_Na_1.7_Ni_5.3_O_82_Sb_2_W_18_	H1299		63.23		[67]
C_25_N_10_Na_1.7_Ni_5.3_O_82_Sb_2_W_18_	HEK293T		114.76		
C_25_N_10_Na_0.7_Ni_5.3_O_76_Sb_2_W_18_	HEK293T		103.09		
C_25_N_10_Na_1.7_Ni_5.3_O_82_Sb_2_W_18_	HepG2		42.98		
PAC-320	LNCaP			5.64	[68]
PW_9_Cu	MC3T3-E1	92.00			[33]
[Co(H_2_O)_6_{CoSb_6_O_4_(H_2_O)_3_[Co(hmta)SbW_8_O_31_]_3_}]^13−^	MCF-7			12.24	[66]
PW_9_Cu	MG-63	22.00			[33]
[Co(H_2_O)_6_{CoSb_6_O_4_(H_2_O)_3_[Co(hmta)SbW_8_O_31_]_3_}]^13−^	OVCAR-3			1.78	[66]
{Sb_9_W_21_}	OVCAR-3			8.80	[67]
[Co(H_2_O)_6_{CoSb_6_O_4_(H_2_O)_3_[Co(hmta)SbW_8_O_31_]_3_}]^13−^	SK-OV-3			15.02	[66]
C_25_N_10_Na_1.7_Ni_5.3_O_82_Sb_2_W_18_	SMMC-7721		48.29		[67]
PW_9_Cu	UMR106	81.00			[33]

* Note: IC_50_ values in the table marked with (*) are measured in µg/mL.

**Table 3 ijms-24-05043-t003:** IC_50_ (µM) of the different antitumor drugs in the respective cell lines.

Compound	Cell Line	IC_50_ µM			Ref.
		24 h	48 h	72 h	
ATRA	HL-60	20.76			[60]
	U937	14.85			
Cisplatin	SH-SY5Y	28.40	11.60		[65]
	MG-63	43.00			[33]
	AGS		17.44		[67]
	BGC-823		5.78		
MTX	HepG2	42.03			[63]
	A549	26.93			
	MCF-7	49.79			
TSA	LNCaP			98.14	[68]
NaB				3.46	
TSA	DU145			59.45	
NaB				1.20	

**Table 4 ijms-24-05043-t004:** Effect of POMs and their cell lines on cell cycle arrest.

Compound	Cell Line	Cell Cycle Arrest	Ref
G1	S	G2/M
POVs
VMOP-31	SMMC-77221		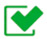	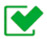	[59]
CoV_10_	SMMC-77221		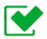		[57]
V_18_	MCF-7			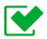	[58]
POMos
K_2_Na[As^III^Mo_6_O_21_(O_2_CCH_2_NH_3_)_3_] 6H_2_O—1	A549		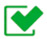		[62]
K_2_Na_2_[γ-Mo_8_O_26_(O_2_CCH_2_NH_3_)_2_]—2	A549		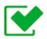	
[{4,4′-H_2_bpy}{4,4′-Hbpy}_2_{H_2_P_2_Mo_5_O_23_}].5H_2_O	A549		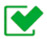		[63]
HepG2		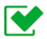	
MCF-7			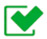
[(Cu(pic)_2_)_2_(Mo_8_O_26_)] 8H_2_O	A549		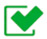		[64]
HepG2		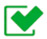	
MCF-7			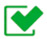
K_2_Na[As^III^Mo_6_O_21_(O_2_CCH_2_NH_3_)_3_] 6H_2_O	HL-60	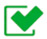			[60]
U937	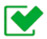		
POM@SiO_2_	MCF-7		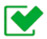		[61]
POPds
Pd_13_	SH-SY5Y		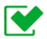		[65]
Pd_13_L		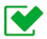	
SrPd_12_		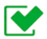	
POTs
[Co(H_2_O)_6_{CoSb_6_O_4_(H_2_O)_3_[Co(hmta)SbW_8_O_31_]_3_}]^13−^ (1)	A2780			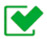	[66]
A2780cis			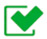
C25N10Na1.7Ni5.3O82Sb2W18—1	AGS		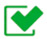		[67]
BGC-823		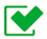	
PAC-320	DU145			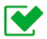	[68]
LNCaP			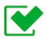
PW_9_Cu	MG-63			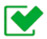	[33]

## Data Availability

The data is available in the original research papers.

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
