# Peer review of "Polyoxometalates Impact as Anticancer Agents"

_ijms, 2023, doi:10.3390/ijms24055043_

Round 1
Reviewer 1 Report
- please consider changing the title for example:
Polyoxometalates impact as anticancer agents
- line 14: ROS write full name at first time of giving abbreviation
please do the following corrections in English mistakes
-line 68: help in fighting
-line63: that (delete one of them)
-line143: found the
-line 177: delete the in the Table
-line 180: referred to the; delete (in)
-line 222: use the instead of this
line 241: delete And
line 418: after interfere with delete at the
from line 379-388: too long for one sentence and you make it a one paragraph; please dived into 3 sentences.
line 266: in vivo study you did not mention any group of healthy animals received polyoxometalates to find out their toxicity compared to reported drugs.
-regarding table 3: you did not discuss the impact of high IC50 values for Polyoxometalates at normal cells compared to cancer cells.
-references: you have to revise carefully name of journals because some times you either write the full name or the abbreviation of the journal.
Author Response
Dear Reviewer (1)
Thank you for the comments and suggestions made to our work, they were certainly very useful to improve it. We appreciate that you did found the topic relevant.
The answers to the comments are described below:
please consider changing the title for example:
Polyoxometalates impact as anticancer agents
In fact, we wish previously to change the title and therefore we totally agree with the reviewer suggestion and the title was changed.
- line 14: ROS write full name at first time of giving abbreviation
please do the following corrections in English mistakes
Corrections indicated below were made and showed in yellow in the revised version:
-line 68: help in fighting
-line63: that (delete one of them)
-line143: found the
-line 177: delete the in the Table
-line 180: referred to the; delete (in)
-line 222: use the instead of this
line 241: delete And
line 418: after interfere with delete at the
from line 379-388: too long for one sentence and you make it a one paragraph; please dived into 3 sentences.
line 266: in vivo study you did not mention any group of healthy animals received polyoxometalates to find out their toxicity compared to reported drugs.
We included the following sentence after the in vivo section studies:
Several studies demonstrate that the oral administration of POMs is presumed safe and poses a low risk of potential health risks. Furthermore, for potential antidiabetic POTs it was concluded that the hepatotoxic and nephrotoxic effects of could be considered as mild. Thus, besides POMs presents higher antitumor activity and lower toxicity in in vitro and in vivo experiments it were also described as promising agents in the treatment of infectious diseases, diabetes and Alzheimer’s disease [new refs: 77,78,79].
- Wang, X., Wei, S., Zhao, C. et al.Promising application of polyoxometalates in the treatment of cancer, infectious diseases and Alzheimer’s disease. J Biol Inorg Chem 27, 405–419 (2022). https://doi.org/10.1007/s00775-022-01942-7
- Qu X, Xu K, Zhao C, Song X, Li J, Li L, Nie W, Bao H, Wang J, Niu F, Li J. Genotoxicity and acute and subchronic toxicity studies of a bioactive polyoxometalate in Wistar rats. BMC Pharmacol Toxicol. 2017 Apr 5;18(1):26. doi: 10.1186/s40360-017-0133-x. PMID: 28381296; PMCID: PMC5382445.
- Dinčić M, Čolović MB, Sarić Matutinović M, Ćetković M, Kravić Stevović T, Mougharbel AS, Todorović J, Ignjatović S, Radosavljević B, Milisavljević M, Kortz U, Krstić DZ. In vivotoxicity evaluation of two polyoxotungstates with potential antidiabetic activity using Wistar rats as a model system. RSC Adv. 2020 Jan 15;10(5):2846-2855. doi: 10.1039/c9ra09790b. PMID: 35496114; PMCID: PMC9048772.
-regarding table 3: you did not discuss the impact of high IC50 values for Polyoxometalates at normal cells compared to cancer cells.
At the end of this section where Table 3 is present we now include the following sentence:
Moreover, taken in consideration the high IC50 values for POMs at normal cells compared to cancer cells it is established that these compounds showed high selectivity towards the cancer cell lines. Therefore, POMs selectively target cancer cells while sparing healthy cells, pointing out as promising agents in the treatment of cancer. In fact, POMs are expected to develop into the next generation of anticancer drugs [15,79].
-references: you have to revise carefully name of journals because some times you either write the full name or the abbreviation of the journal.
Abbreviations of the journals were corrected and indicated the correction in yellow and in track changes in the revised version.
We would like to once again express our thanks to the Reviewer for revising this review and for their valuable comments which helped us to improve this manuscript. We tried our best to address every Reviewer’s comment as precisely and carefully as possible. We took every comment very seriously and hope that we were able to remove major flaws and uncertainties/doubts to the satisfaction of the Reviewer and Editor. Considering the improvements and clarifications, we would highly appreciate acceptance for publication in IJMS of this revised and resubmitted manuscript. MA 19/02/2023

Reviewer 2 Report
The article entitled “Polyoxometalates effects in cancer: A Systemic Review” concerns an interesting class of compounds which, thanks to their wide range of applications, have been the subject of interest of many research centers in recent years, which is confirmed by the number of publications appearing on them. However, the method of presenting selected literature data chosen by the authors, in my opinion, is devoid of any deeper thought and constitutes a statistical collection of data, which is well illustrated in Figures 3 and 4.
I do not fully understand the point of including the Methodology chapter in the article. At this scientific level, the rules of conducting literature searches on the Internet are usually well known and I believe that the description provided does not bring anything significant to the work.
The bibliographic data collected by the authors may be considered interesting, but in my opinion they do not fit well as a scientific article in the trend presented by the International Journal of Molecular Sciences. The nature of the article is best reflected in the sentence placed by the authors at the beginning of the Conclusions chapter: In this review paper, a bibliographic research was carried out with the keywords “Polyoxometalates” AND “cell cycle”. Therefore, it is difficult to consider it as a comprehensive review, and therefore I do not recommend this article for publication in IJMS.
Author Response
Dear Reviewer
Thank you for the comments made to our work. We appreciate that you did found the topic relevant.
The research group responsible for the review paper authors do have experience in the field and also in review papers as well chapters in the field. Some examples can be found at references 10,15,16,18,19,21,22,23,25,26. However, they followed several patterns and structures due the specific goals of each review. Thus, they are not similar and can be distinguish from each other. In the present review we also try to change the style as possible from the recent reviews published in Metals, Biochem and CCR for example.
In the case of the review published in Metals, the Methods section that was included, it was based in a reviewer suggestion. We did agree to include a methodology section and a schematic figure resuming the criteria used for the bibliographic research. We thought that improved the paper. In fact, this publication entitled Vanadium in melanoma: a systematic review” received so far 20 citations since 2021. Taken in consideration this very good response from the scientific community in the field, it was decided to followed the same pattern including a methodology section. Besides the schematic clarification of the selection criteria and why some papers were not included, it is useful for the global understanding of the paper, and also pedagogical for the researchers and students.
But we will only know after publication, if the review is timely and interesting for the community working in this specific interdisciplinary field. So far, we have a relative very good number of citations with the review referred above in Metals, and also recently for the Biochem and CCR reviews. Finally, the title was changed as suggested also by the first reviewer in order to best reflect the review: “Polyoxometalates impact as anticancer agents”.
We tried our best to address every Reviewer’s comment as precisely and carefully as possible. We took every comment very seriously and hope that we were able to remove major flaws and uncertainties/doubts to the satisfaction of the Reviewer and Editor. Considering the improvements and clarifications, we would highly appreciate acceptance for publication in IJMS of this revised and resubmitted manuscript.
MA 19/02/2023

Round 2
Reviewer 2 Report
In the revised version of the article, the authors changed the title, but the main part of the text remained unchanged. The changes introduced in the text improved its value, but did not change the approach to the literature review. Authors certainly do not lack experience in writing review articles, which can be confirmed by the self-citations contained in the text of this article. However, when it comes to the compliance of this type of review article with the publishing policy of IJMS, it still raises my doubts.
Author Response
The methodology part was removed from the review. Changes in yellow were introduced according to reviewers suggestions.